# Endophytic Fungus Negatively Affects Salt Tolerance of Tall Fescue

**DOI:** 10.3390/jof9010014

**Published:** 2022-12-21

**Authors:** Aino Kalske, Kari Saikkonen, Marjo Helander

**Affiliations:** 1Department of Biology, University of Turku, 20014 Turku, Finland; 2Biodiversity Unit, University of Turku, 20014 Turku, Finland

**Keywords:** endophyte, salinity, competition, *Epichloë*, symbiosis, fungi

## Abstract

Vertically transmitted endophytic fungi can mitigate the negative effects of salinity encountered by their host grass and alter the competitive interactions between plant individuals. To experimentally study the interactive effects of the fungal endophyte *Epichloë coenophiala* on salt tolerance and intraspecific competition of its host plant, tall fescue *Festuca arundinacea*, we subjected 15 maternal lines of each *Epichloë* associated (E+) and *Epichloë* free (E−) tall fescue to salt treatment and competition in the greenhouse and common garden. Then, to explore variation in endophyte incidence in natural populations of tall fescue, we surveyed 23 natural populations occurring on or near the Baltic Sea coast in Aland islands in southwestern Finland for endophyte incidence, distance to shore, and competitive environment. Under salinity in the greenhouse, E− plants grew larger than E+ plants, but there was no size difference in the control treatment. E− plants grew taller and were more likely to flower than E+ plants when grown in benign conditions in the common garden but not with salinity or competition. The frequency of *Epichloë* incidence was high (90%) in natural populations, and it decreased towards the shore and risk of salt exposure. These results demonstrate a negative effect of *Epichloë* endophyte on the salt tolerance of its host. The high incidence of *Epichloë* in natural populations of tall fescue in the northern part of the species distribution range is likely due to factors other than salinity.

## 1. Introduction

Soil salinity due to both natural and human-induced causes is emerging as one of the key abiotic stressors limiting agricultural productivity worldwide [1]. High concentrations of salts in soils stress plants by hindering water uptake and through immediate effects on cell growth [2]. Salt stress typically causes reduced root growth, faster leaf senescence, and reduced photosynthetic capacity [3]. Strategies to tolerate high salinity include changes in growth, physiology, metabolic pathways, and gene expression [2,4]. Exploring the factors that allow plants to cope with salinity could be a useful approach to overcoming the challenges of increased soil salinity [5].

Plant-associated microbes can mitigate the effects of various biotic and abiotic stressors in their host plants [6,7,8]. For example, some plant-associated fungi can improve the salt tolerance of their host plant via an increase in total biomass, alteration of root architecture, changes in osmoregulation, increased capacity to tolerate oxidative stress, and improved nutrient uptake (reviewed in [8]). A special case of plant-associated fungi are the systemic, vertically transmitted (hereditary via seeds), often asexual symbiotic endophytes that associate with many cool-season grass species. These endophytic fungi are asymptomatic, entirely dependent on the host’s resources, and form a life-long symbiosis in the intracellular space of the above-ground parts of their host plant [9,10]. Although not inhabiting the root space, there is some limited evidence that these systemic endophytes may alter the salt tolerance of their host through anatomical changes [11] or via interactions with other plant microbes [12].

Because systemic endophytes tend to have a positive effect on growth, reproduction, and resistance to abiotic and biotic stressors of their hosts [9,13,14,15,16], endophyte association has commonly been suggested to improve competitive abilities compared to endophyte-free plants [17,18] (but see [19]). However, this benefit may be context-dependent, as the competitive benefit provided by the endophyte can change depending on herbivory, moisture, and nutrient availability [6,13,14,20,21]. As the energetic costs of salt tolerance can be substantial [22], it too has the potential to alter the competitive benefit conferred by the endophyte to its host plant. Thus far, the effects of endophyte association on the salt tolerance of the host have been explored without considering such ecological interactions.

Despite their many reported benefits to the host plant, frequencies of endophyte symbionts are variable across the natural populations of their hosts [23,24,25,26]. In the case of vertically transmitted endophyte symbionts, their frequency should be proportional to the benefit they confer on their host [27,28,29,30]. These benefits may be contingent upon the environment: the frequency of endophyte association increased with herbivory in the grass-tall fescue in the U.S. [31]. In the halophytic grass, *Puccinellia distans,* the frequency of horizontally transmitted endophyte-hosting plants in a population was negatively correlated with salinity [32]. If endophyte association confers salt tolerance to its host, and this is translated to differences in fitness under salt exposure, salinity can influence the distribution and frequency of the endophyte incidence across populations.

*Epichloë coenophiala* (Morgan-Jones & W. Gams) C.W. Bacon & Schardl [formerly *Neotyphodium coenophialum* (Morgan-Jones & W. Gams) Glenn, C.W. Bacon & Hanlin) is an asexual, systemic endophytic fungus associating with the perennial grass tall fescue (*Festuca arundinacea,* Schreb. [syn. *Schedonorus phoenix* (Scop.) Holub. and *Lolium arundinaceum* (Schreb.) Darbysh.]). This fungus is strictly maternally vertically transmitted and is considered typically to be a mutualist, especially in nutrient-rich agroenvironments [9,28]. When exposed to high salinity, endophyte association increased leaf survival, reduced the accumulation of sodium (Na^+^) and chloride (Cl^-^) in root tissues [33], and increased accumulation of Na^+^ in leaf and sheath tissue of their tall fescue hosts [34]. To what extent these effects translate to differential growth, survival, and fitness of the host is still unclear. *Epichloë* association also improves the competitive ability of tall fescue in its exotic range in a species-diverse community [17,35] as well as in intraspecific competition [36]. However, in its native range in Eurasia, the species is not as competitively dominant as in the introduced range, despite high endophyte association frequency [19,23,24]. Given that the endophyte may potentially affect both salt tolerance and the competitive ability of its host, the interactive effects of salt and competition can have repercussions on population endophyte frequency and plant community dynamics. 

We have begun to understand the effects of vertically transmitted plant endophytes on the salt tolerance of their host grasses. To date, these efforts have largely focused on a limited number of genotypes and/or ecologically unrealistic conditions [33,34]. Furthermore, whether the effects of systemic endophytes on salt tolerance of the host influence the endophyte frequency in natural populations is virtually unexplored. We used wild-collected maternal lines of tall fescue in their native range in Europe with known *E. coenophiala* status (with and without endophyte; E+ and E−, respectively) to answer the following questions: 1. Does endophyte association affect the salt tolerance of tall fescue? 2. Does salt treatment alter the competitive dynamics between plants of different endophyte statuses? and 3. Is the probability of endophyte association related to salt exposure and/or competition in natural populations? We measured plant growth, photosynthetic capacity, survival, and reproduction in the greenhouse and common garden experiments. We predicted that endophyte association would increase the salt tolerance of tall fescue, demonstrated by the increased growth and survival of E+ plants compared to E− plants. We also predicted that the E+ plants would be better competitors than E− plants and that this difference would be augmented in salt treatment. Finally, we predicted that in natural populations of tall fescue, E+ plants would grow closer to the seashore with increasing salt exposure than E− plants.

## 2. Materials and Methods

Tall fescue *Festuca arundinacea* (Poaceae) is a perennial grass native to Europe. In the north, its range extends to southern and southwestern Finland, where the species grows on its range limit. In Finland, it typically grows along seashores on the brackish Baltic Sea coast and along roads further inland [23]. In the seashore populations, the species grows in rock crevices with low nutrient availability and high exposure to brackish seawater. The individuals closest to the shore are often found less than one meter from the waterline and are continuously exposed to seawater. The Baltic Sea does not have tidal variation in sea level, but strong winds and waves cause the effects of seawater to extend further up the shore. The average salinity of the seawater in the study area is 6.5 ppt (parts per thousand) [37]. Tall fescue has an association with the systemic vertically transmitted (maternally from parent to offspring) endophytic fungus *Epichloë coenophiala* (Ascomycota: Clavicipitaceae). This fungal endophyte lives asymptomatically and internally throughout the aboveground parts of its host plant, including the seeds, thus transmitting vertically from the maternal plant to the offspring.

### 2.1. Experimental Setup

We used seeds from 30 maternal plants (15 E+ and 15 E−) growing in a common garden established in 2016. Plants in the common garden originated from the Aland islands in southwestern Finland, and their endophyte status was determined by microscopy (up to three seeds per maternal plant). The presence of the systemic *Epichloë* endophyte hyphae in the seed is an indication of endophyte association [38]. We collected seeds from the common garden experiment in 2018 and stored them at +4 °C until use. We sowed seeds in a greenhouse with ambient light and temperature in late April 2020 in plug trays with a standard potting mix with 3–5 seeds per plug. At the end of May, we moved seedlings to plugs individually with Kekkilä amppeliseos potting mix, where they grew for one week prior to the start of the experiments.

#### 2.1.1. Greenhouse

To explore how endophyte association affects the salt tolerance of tall fescue and whether competitive interactions are altered due to salt treatment, we established a fully factorial greenhouse experiment with endophyte (E−, E+), salt treatment (yes/no) and competition (yes/no) treatments (8 treatment combinations). Each maternal plant (15 E+ and 15 E−) was used three times, and each treatment combination was, thus, repeated 45 times, resulting in 360 plants (3 × 15 maternal plants × 8 treatments). At the start of the experiment at the beginning of June 2020, we moved seedlings to 1 L pots filled with sand. Plants in the competition treatment had two plants per pot, and plants without competition were in pots individually. All competitors were E+ because in the wild tall fescue populations on Aland islands >90% of the plants harbor the *Epichloë* endophyte [23,24], and intraspecific competition would, thus, typically be more likely to occur against an endophyte symbiotic plant. The maternal origin of competitors was randomly assigned, and the orientation of the two plants in the competition pots was chosen haphazardly. We arranged the pots in three randomized blocks with one individual per maternal line in each of the salt and competition treatments in each block. We started the salt treatment three days after planting. Salt water-treated plants received 100 mL of salt water with 7 ppt (0.7%) concentration (7 g NaCl/1 L water) which is equivalent to the average salinity of the Baltic Sea. Water control plants received an equivalent amount of tap water. Salt watering treatment was repeated three times a week for 12 weeks until the end of the experiment. Plants received no other water.

We measured the initial size of the focal plants and competitors (height of the tallest leaf and number of tillers) in early June. To estimate the photosynthetic capacity of the plants, we measured chlorophyll content with a portable chlorophyll meter (SPAD-502 Plus, Konica Minolta) in July by taking three readings from one average-sized, fully expanded leaf blade per plant. The SPAD value (arbitrary unit) provides an indication of the relative amount of chlorophyll present in the leaf. Higher values generally indicate healthier plants. We terminated the experiment in September and harvested all the shoots and roots separately. We dried the plant material at 60 °C for 72 h prior to weighing them and calculated root:shoot ratio by dividing root mass by shoot mass. 

#### 2.1.2. Common Garden

To explore the interactive effects of endophyte, salt treatment, and competition in field conditions, we established a similar fully factorial common garden experiment as in the greenhouse with endophyte status, salt treatment, and competition, but with only one replicate per maternal plant line per treatment (8 × 15 = 120 plants). At the beginning of June 2020, we planted focal plants in a fenced sand field in eight rows, with 75 cm between rows and 65 cm between plants within rows. The field was cleared of any other plants prior to the experiment, and weeding was repeated as necessary to avoid weed proliferation. Every other plant in each row was subject to salt treatment resulting in a checkered pattern; the other treatments were randomly distributed. Plants in the competition treatment had two competitors on opposite sides of the plant at an 8 cm distance from the focal plant. Watering and salt treatment were administered with a drip watering system to the focal plant starting the day after planting. Each plant in the salt treatment received 500 mL of 7 ppt (0.7%) salt water, and those in the water control treatment an equal amount of water. We repeated watering/salt treatment three times a week until the end of August 2020 and continued it again in 2021 from the beginning of June to the end of August (12 weeks each year). Plants were exposed to rain in addition to our watering treatment. At the end of August 2020, we harvested all the aboveground tissue and weighed it after drying it at 60 °C for 72 h. In 2021, we measured plant height once in the early season and recorded plant survival and plant stage (vegetative or flowering) in August. We harvested and weighed all aboveground biomass in 2020. 

### 2.2. Field Populations

To explore the infestation rates of tall fescue in natural populations in relation to salt exposure and competition, we conducted field measurements in 23 natural populations in the Aland islands in August 2020. Fourteen of the populations were located on the seashore, 8 along roads, and one in a pasture. In each population, we recorded data on 10 randomly chosen reproductive tall fescue individuals (target plants). By measuring reproductive individuals, we directed our focus on plants that were well adapted to the prevailing environmental conditions, as demonstrated by their survival to the reproductive stage. We measured the cover of tall fescue (conspecifics) and all other plants (heterospecifics) within a 50 cm radius and the distance to the waterline of the seashore (when under 10 m) from the target plant. Plants further away than 10 m had a distance to shore value of 10 (103 out of 230 plants). We collected seeds from all the target plants and determined their endophyte status by microscopy (up to three seeds per maternal plant) for the presence of the systemic *Epichloë* endophyte hyphae [38]. 

### 2.3. Statistical Analysis 

Statistical analyses were conducted in R version 4.0.3 [39].

#### 2.3.1. Greenhouse and Common Garden

To analyze the effects of endophyte status, salt treatment, and competition on plant performance, we used linear mixed models (LMM using lmerTest::lmer; [40] and generalized linear mixed models (GLMM using lme4::glmer; [41]). We conducted LMMs for total biomass, root:shoot ratio, and SPAD-values for greenhouse plants with endophyte status, salt treatment, competition, and their interactions as fixed explanatory variables and maternal plant and block as random explanatory variables. For common garden plants, we conducted LMMs for 1st year shoot biomass, 2nd year height, and shoot biomass, with the same fixed explanatory variables as in the previous models and maternal plant as a random explanatory variable. Total biomass, root:shoot ratio and 2nd year shoot biomass were log-transformed and 1st year shoot biomass square-root transformed to meet the assumptions of normality. We determined the significance of the fixed variables using *F*-tests with the Kenward-Roger method for adjusting the denominator degrees of freedom (lmerTest::anova). We analyzed survival and flowering in August of the 2nd year in the common garden with GLMMs using the same explanatory variables as in the previous models. We specified a binomial distribution and a logit link function for both variables. We determined the significance of the fixed variables for GLMMs using type II Wald’s Χ2 test (car::Anova). We assessed pairwise differences in mean values with a Tukey’s test when necessary (emmeans::emmeans; [42]).

#### 2.3.2. Field Populations

We analyzed the effects of salt exposure and competition on plant endophyte status with a GLMM (lme4::glmer; [41]). We had endophyte status (E−, E+) as the response variable and distance to shore and coverage of conspecifics and heterospecifics as fixed explanatory variables, and population as a random variable. Distances beyond 10 m were all recorded as 10 m. We specified a binomial distribution and a logit link function for the model. We determined the significance of the fixed variables for GLMMs using type II Wald’s Χ2 test (car::Anova).

## 3. Results

### 3.1. Greenhouse

All treatments affected the plant performance either independently or via an interaction (Table 1). The effect of endophyte status on plant biomass was modified by salt treatment (Table 1 and Figure 1a). E− plants were 30% larger than E+ plants in terms of total biomass in the salt treatment, but there was no difference between E+ and E− in the water treatment, where both grew larger than in the salt treatment (Figure 1a). The effect of salt treatment was modified by competition in the case of total biomass and root:shoot ratio (Table 1 and Figure 1b,c). Although plants grew larger when growing individually than in competition in both watering treatments, the difference between plants with and without competition was smaller in salt treatment than in control: plants without competition were larger than those in competition by 26% in salt treatment and by 77% in water control. Root:shoot ratio was not affected by competition in the water control treatment, but in salt treatment, plants with competition allocated 28% more to roots than plants without competition. E+ plants had a higher root:shoot ratio than E− plants (1.03 ± 0.05 and 0.88 ± 0.04, respectively; emmean ± SE). Chlorophyll content (SPAD) was higher in E+ than E− (E+ 30.0 ± 0.9; E− 25.3 ± 0.9), in salt treatment than in water control (salt 32.9 ± 0.8; water 22.3 ± 0.8) and in plants growing individually than those with competition (without competition 29.2 ± 0.8; with competition 26.1 ± 0.8). 

### 3.2. Common Garden

Salt treatment affected all plant performance traits either independently or through interaction in the common garden (Table 2). Plants in salt treatment were smaller in both years in terms of aboveground biomass (1st year salt 0.84 ± 0.09 g; water 1.11 ± 0.10; 2nd year salt 0.99 ± 0.24; water 1.94 ± 0.37 g) and less likely to survive (probability of survival to end of 2nd year salt 0.62 ± 0.06; water 0.91 ± 0.04). Height in 2nd year and flowering probability were affected by the three-way interaction of endophyte status × salt × competition (Table 2 and Figure 2). E− plants grew 63% taller and were 178% more likely to flower than E+ plants when grown in benign conditions in control water treatment without competition, but not in other treatment combinations (Figure 2a,b). Salt treatment affected E+ plant height negatively in competition (−31%) but not without it, whereas E− plant height was not affected by salt treatment either in competition or without it (Figure 2a). Competition increased E+ plant height in water treatment but not when exposed to salinity (Figure 2a). E− plants were more likely to flower without competition than with competition when in control water treatment (Figure 2b).

### 3.3. Field Populations

Of the sampled 230 field plants, only 24 were endophyte-free (endophyte incidence was 90%). E− plants were detected in 12 out of 23 populations, and occurrence ranged from 1 to 4 individuals per population. Therefore, all populations consisted predominantly of endophyte-infected plants. Distance to shore predicted endophyte status with E− plants growing on average closer to shore and to salt exposure (*X^2^* = 5.29, df = 1, *p* = 0.023). The average distance to shore across all plants for E− was 3.54 ± 0.79 m and 5.82 ± 0.30 m for E+. 21% of E− and 48% of E+ plants were further than 10 m away from the shore, and the average distance to the shoreline for plants within 10 m was 1.85 ± 0.49 for E− and 1.98 ± 0.21 for E+. Competitive environment did not affect endophyte status (conspecifics: *X^2^* = 2.38, df = 1, *p* = 0.123; heterospecifics: *X^2^* = 0.33, df = 1, *p* = 0.566). 

## 4. Discussion

Salt treatment had clear negative effects on the growth, photosynthetic capacity, and survival of the tall fescues in the greenhouse and in the common garden. Unexpectedly, E− plants were better at tolerating salt stress than E+ plants both in our experiments and in natural populations. E− grew larger than E+ in salt treatment but not in water treatment in the greenhouse, and the height of E− was not affected by salt treatment, unlike that of E+, that were smaller in salt treatment than in water treatment when in competition. In the greenhouse, the allocation of resources was shifted towards shoots in salt-treated plants and in E− plants compared to E+ plants. The competition had a smaller effect on plant performance than salt treatment, but nonetheless caused plants to allocate relatively more resources to roots than plants without competition, although only in the salt treatment. E− plants were taller and more likely to flower than E+ in benign conditions. The distribution of E− plants in natural populations is in line with our experimental results: E− plants grew on average closer to shoreline and salt exposure than E+ plants. Despite the susceptibility to salt stress, endophyte association was ubiquitous across all studied tall fescue populations. Our results clearly demonstrate that *Epichloë* endophyte does not contribute to the salt tolerance of its host, the tall fescue. 

Salinity already at the moderate 0.7% level was indisputably a stressor to tall fescue as it led to reduced size and survival of the plants. Opposite to our predictions, endophyte association did not improve the salt tolerance of tall fescue in our experiment. In fact, the reverse was the case for some of the measured traits, where E− plants outperformed E+ plants in saline conditions. Endophytes in the *Epichloë* genus have been linked to physiological changes in response to salt treatment, and these have sometimes been linked to improved plant performance. Sabzalian & Mirlohi [33] reported that one out of two tested genotypes had higher leaf survival in 1% salinity treatment when associating with an endophyte compared to being endophyte-free, but they did not find an effect of endophyte on any of the other measured plant performance or growth traits. Yin et al. [34] reported a higher accumulation of Na^+^ in aboveground tissues in E+ than E− plants in a hydroponic system with 1.5% salinity, but this difference was not clearly translated to differential growth. Pereira et al. [12] found a higher accumulation of Na^+^ in leaves of E− than E+ in red fescue (*Festuca rubra* subsp. *pruinose*) in 3.5% salinity, but similarly, no effects on plant growth. We found differential effects of the endophyte on plant growth at lower salinity than what previous studies used, highlighting the need for the use of more genotypes or maternal lines as well as experiments in more natural conditions in further studies. 

E+ plants had a higher root:shoot ratio than E− plants, indicating a higher allocation of resources to roots than shoots. Because one of the two main ways in which salinity affects plants is through osmotic stress and reduced water availability due to salts outside the roots [2], higher root:shoot ratios have been suggested to be linked to better salt tolerance [43]. This was not the case in our study. Similar to our results, a higher root:shoot ratio did not predict salt tolerance in different cultivars of quinoa (*Chenopodium quinoa*) [44] or among several wheat varieties (*Triticum* sp. and *Aegilops* sp.) [45]. It is possible that larger root systems favor the higher accumulation of ions and greater sensitivity to the ionic stress created by Na^+^ accumulation in the leaves.

The susceptibility to salt caused by endophyte association in tall fescue is in accordance with field data where E− plants were, on average, closer to the seashore and salt exposure than E+ plants. Despite their susceptibility to salinity, the frequency of E+ plants in the populations were high overall (90%), which is in line with previous results [24]. The high prevalence of symbiosis suggests that it must convey benefits that outweigh the cost to the host plant in order to persist in the population as purely a vertically transmitted endophyte [16,28,30]. Factors that have typically been used to explain the maintenance of the endophyte in plant populations include improved growth, herbivore resistance, drought tolerance [18], as well as metapopulation dynamics combined with effective seed dispersal [46].

According to previous research, *Epichloë* symbioses improve the competitive ability of tall fescue [17,35]. Although we found a negative effect of competition on plant performance overall in terms of biomass and photosynthetic capacity (SPAD), our results do not demonstrate a competitive advantage provided by the endophyte. In the common garden, competition reduced flowering in E− only, but this was due to a lower flowering probability of E+ compared to E− in the absence of competition. Similarly, E+ grew taller in competition than without it, but likewise, this did not result in E+ being taller than E− in competition. Differences in height may have been due to a more sprawling growth habit under benign conditions, as we did not see comparable effects on plant biomass. Previous studies that demonstrate improved competitive ability owing to endophyte association in tall fescue were performed in the invasive range of the species and, thus, may not be relevant in the case of the native populations [19]. Similar to our results, the *Epichloë* association reduced the competitive abilities of Arizona fescue (*Festuca arizonica*) in its native range [47].

In addition to differences among plant origins, the competitive benefit may be contingent upon the abiotic and biotic environment. Some of the benefits from the endophyte association only manifest in nutrient-rich conditions [9,28]. We used a low nutrient growth medium to match the conditions in natural populations, which could have affected the outcome of the competitive interactions. Furthermore, in our study, the competition was against conspecifics, and some of the competitive advantages may only be evident in competition against heterospecifics [17]. Nonetheless, the competitive environment did not explain the incidence of endophyte symbiosis in the natural populations and may not be the cause for the high prevalence of *Epichloë* in the study area. 

Association with *Epichloë* provides its host tall fescue with increased protection against herbivory by the production of ergot alkaloids and lolines [48]. Such benefit could be a reason for the high incidence of *Epichloë* in the natural populations in the study area [49]. Alternatively, endophytes can benefit their host in tolerating other abiotic stress, such as waterlogging [50]. This could be beneficial in our natural populations as the changes in water level cause some of the plants to be regularly exposed to waterlogged conditions. Cold tolerance can be another abiotic stress that would be relevant for the survival of the grass in the northern part of its range, but at least in the invasive range, endophyte association did not affect the cold tolerance of tall fescue [51]. However, the benefits from the symbiosis that warrant the high incidence in natural populations in our study area remain to be solved in future studies.

This study elucidates the negative effect of the symbiotic endophytic fungi on the salt tolerance of tall fescue. Previous studies conducted with limited numbers of plant genotypes have found physiological effects of the *Epichloë* endophyte on their host grass without marked effects on plant growth. While we did not explore the physiological mechanisms behind the differences in salt tolerance, our experimental and field data clearly point to increased susceptibility to salinity in *Epichloë* associated tall fescue. The negative effect of the endophyte is strong enough that it affects the distribution of the symbiont-free plants in our study populations experiencing salt stress and may even contribute to the maintenance of variation in incidence rates in natural populations in general.

## Figures and Tables

**Figure 1 jof-09-00014-f001:**
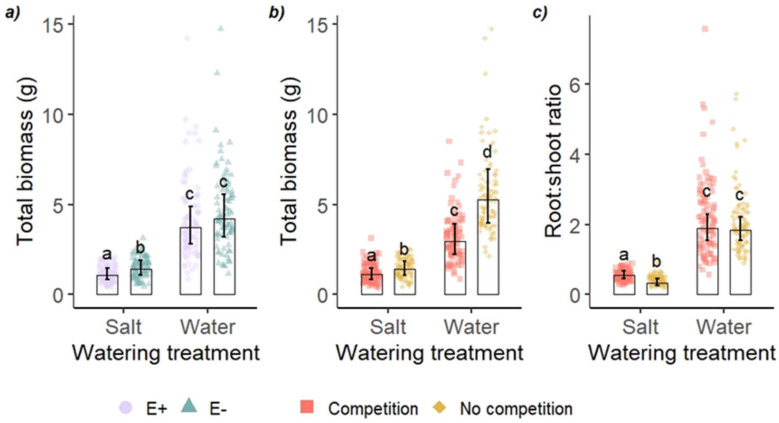
Total biomass and root:shoot ratios of tall fescue (*Festuca arundinacea*) grown in a greenhouse, exposed to salt watering treatment (**a**) with (E+) and without (E−) endophytic fungus, and (**b**), (**c**) with and without competition. Bars represent estimated marginal means (± 95% confidence limits), back-transformed. Points are individual plants. Bars that do not share a letter within a panel are different from one another (Tukey’s test). Legend in panel **b** is the same for panel **c**.

**Figure 2 jof-09-00014-f002:**
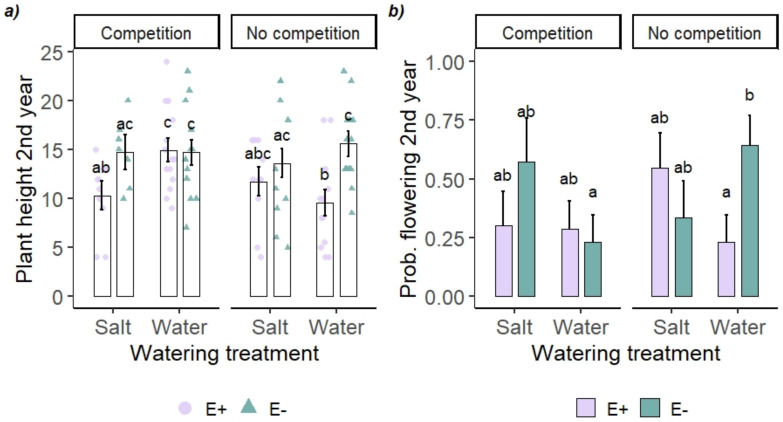
Height (**a**) and probability of flowering (**b**) of endophyte symbiotic (E+) and endophyte-free (E−) tall fescue (*Festuca arundinacea*) exposed to salt watering treatment and competition in a common garden. Bars represent estimated marginal means (**a**) and probabilities (**b**) ± standard errors. Points in panel (**a**) are individual plants. Bars that do not share a letter within a panel are different from one another (Tukey’s test).

**Table 1 jof-09-00014-t001:** Results from mixed models for the effects of endophyte status, salt treatment, and competition on performance of tall fescue *Festuca arundinaceae* in the greenhouse. Maternal plants and blocks were included as random factors in all models. df and ddf denote the degrees of freedom in the numerator and denominator, respectively. Transformations (if any) in parenthesis after the variable name.

	Total Biomass (log)		Root:Shoot Ratio (log)		Chlorophyll Content	
**Explanatory Variables**	* **F** * _ **df,ddf** _	* **p** *	* **F** * _ **df,ddf** _	* **p** *	* **F** * _ **df,ddf** _	* **p** *
Endophyte (E)	5.35_1,28_	**0.028**	11.45_1,28_	**0.002**	35.47_1,28_	**<0.001**
Salt (S)	1115.93_1,322_	**<0.001**	1472.58_1,322_	**<0.001**	582.12_1,322_	**<0.001**
Competition (C)	135.16_1,322_	**<0.001**	10.71_1,322_	**0.001**	50.30_1,322_	**<0.001**
E × S	3.99_1,322_	**0.047**	1.40_1,322_	0.238	0.03_1,322_	0.870
E × C	0.02_1,322_	0.878	2.03_1,322_	0.155	0.00_1,322_	0.973
S × C	24.85_1,322_	**<0.001**	10.57_1,322_	**0.001**	0.18_1,322_	0.676
E × S × C	0.10_1,322_	0.752	0.05_1,322_	0.816	0.14_1,322_	0.711

*p*-values <0.05 in **bold**.

**Table 2 jof-09-00014-t002:** Results from mixed models for the effects of endophyte status, salt treatment, and competition on performance of tall fescue *Festuca arundinaceae* in the common garden. The maternal plant was included as random factor in all models. df and ddf denote the degrees of freedom in the numerator and denominator in LMMs, respectively (for GLMMs df is one). Transformations (if any) in parenthesis after the variable name.

	Shoot Biomass 1st year (sqrt)		Height June 2nd year		Shoot Biomass 2nd year (log)		Survival		Flowering Probability	
**Explanatory Variables**	* **F** * _ **df,ddf** _	* **p** *	* **F** * _ **df,ddf** _	* **p** *	* **F** * _ **df,ddf** _	* **p** *	* **X** ^ **2** ^ *	* **p** *	* **X** ^ **2** ^ *	* **p** *
Endophyte (E)	0.26_1,28_	0.616	9.10_1,25_	**0.006**	0.01_1,27_	0.921	1.29	0.256	0.94	0.333
Salt (S)	5.44_1,82_	**0.022**	1.21_1,69_	0.274	5.03_1,70_	**0.028**	10.89	**0.001**	0.62	0.431
Competition (C)	1.70_1,82_	0.196	1.11_1,64_	0.297	0.80_1,64_	0.375	0.48	0.489	1.17	0.279
E × S	0.18_1,82_	0.671	0.02_1,69_	0.898	0.66_1,70_	0.419	0.45	0.503	0.67	0.413
E × C	1.67_1,82_	0.200	0.88_1,64_	0.352	0.80_1,64_	0.374	0.37	0.543	0.09	0.770
S × C	0.93_1,82_	0.337	1.42_1,68_	0.237	0.95_1,69_	0.332	0.15	0.695	0.43	0.512
E × S × C	3.43_1,82_	0.068	4.80_1,68_	**0.032**	1.44_1,69_	0.234	0.39	0.535	4.84	**0.028**

*p*-values <0.05 in **bold**.

## Data Availability

The data presented in this study are available on request from the corresponding author.

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
