# Peer review of "Endophytic Fungus Negatively Affects Salt Tolerance of Tall Fescue"

_jof, 2022, doi:10.3390/jof9010014_

Round 1

Reviewer 1 Report

This paper is well-written and the research presented is scientifically sound.  One comment is that while I understand that the competition trials only make sense for E+ due to natural occurrences, it would be interesting to see the difference in competition with E+ and E- .  Perhaps in a future study. 

I think you could clarify the figure two legend.  It states total biomass and root:shoot ratios of TF.... but the actual figures show Plant height 2nd year and Prop flowering 2nd year.  This is confusing to the reader.

Author Response

Thank you for your supportive comments. We agree that it would be interesting to further investigate the competitive interactions between E+ and E- by a fully crossed competition study.

We are very grateful for you for noticing the error in the figure 2 legend. We had included the figure 1 legend for both figures in the initial submission by mistake. We have fixed this, and the correct legend now reads as follows: “Height and probability of flowering of endophyte symbiotic (E+) and endophyte free (E-) tall fescue (Festuca arundinacea) exposed to salt watering treatment and competition in a common garden. Bars represent estimated marginal means (± standard errors), back-transformed. Points in panel a) are individual plants. Bars that do not share a letter within a panel are different from one another (Tukey’s test).”

Reviewer 2 Report

The authors should be commended for a thorough and well-executed study that makes an important contribution to the field of plant-endophyte interactions. The combination of a greenhouse experiment, a common garden experiment and field observations from natural populations makes a very convincing case that Epichloe endophytes – widely regarded as mutualistic symbionts that increase stress tolerance in their hosts, particularly in tall fescue – have a negative effect on salt tolerance of their hosts.  This finding is all the more significant because it contrasts with the findings of previous studies that had a more physiological focus.

68 – “…endophytic fungus associating with…”

84-5 – “…on a limited number…”

87 – “…in natural populations is…”

89 – “…to answer the following…”

122 – “…stored them at +4…”

126 – “…prior to the start of…”

147 – was this the only water given?

151-2 – was the age of the leaf used consistent?  If so, indicate which leaf (youngest fully expanded, etc.)

163 – “…focal plants in a fenced sand field…”

163 – was there no other vegetation in the field? How was this maintained if so

174 – specify “plant height” here (rather than size)

181 – “Fourteen of the populations…”

182 – were the roadside populations natural or planted?

183 – did the use of only reproductive plants introduce a bias?

185-6 – what proportion of the plants were >10m from the shoreline?

214 – replace “treatment” with “exposure” here?

268-70 and 352-4 – height may not be a reliable indicator of overall plant size or performance when comparing plants with and without competition, since the light environment can affect growth habit.  Interestingly it appears there may have been an interaction of endophyte infection and plant growth habit.  Figure 2 shows that  endophyte-infected plants were actually taller under competition (without salt) than they were under benign conditions, whereas endophyte-free plants were similarly tall under both conditions.  This suggests that endophytes were inducing a more sprawling growth habit under benign conditions (i.e., in the open).  Endophytes also affected chlorophyll content, supporting the possibility that they may alter plant response to light.

290-91 – If many plants were >10m from shore, average distance to shore may not be the most meaningful measure to use here.  Should these data be presented as “> 10 m” vs “< 10 m”, with average distance calculated for the second group?

304 – “…only in the salt…”

347 – “…symbioses improve the…”

Author Response

Reviewer 2

C: The authors should be commended for a thorough and well-executed study that makes an important contribution to the field of plant-endophyte interactions. The combination of a greenhouse experiment, a common garden experiment and field observations from natural populations makes a very convincing case that Epichloe endophytes – widely regarded as mutualistic symbionts that increase stress tolerance in their hosts, particularly in tall fescue – have a negative effect on salt tolerance of their hosts.  This finding is all the more significant because it contrasts with the findings of previous studies that had a more physiological focus.

Re: Thank you very much for these encouraging words. We have implemented all the textual changes suggested below as well as addressed the other comments (see our specific responses below).

C: 68 – “…endophytic fungus associating with…”
C: 84-5 – “…on a limited number…”
C: 87 – “…in natural populations is…”
C: 89 – “…to answer the following…”
C: 122 – “…stored them at +4…”
C: 126 – “…prior to the start of…”

Re: We have done all these textual changes suggested above.

C: 147 – was this the only water given?

Re: Yes, the plants received no other water in the greenhouse. In the common garden they were exposed and received rain water in addition to our watering treatment. We now mention this in the manuscript (lines 148-149 and 175-176).

C: 151-2 – was the age of the leaf used consistent?  If so, indicate which leaf (youngest fully expanded, etc.)

Re: The age of the leaf was not controlled for.

C: 163 – “…focal plants in a fenced sand field…”
Re: Done

C: 163 – was there no other vegetation in the field? How was this maintained if so

Re:There was no other vegetation. Common garden was weeded as needed to prevent weed proliferation. We now mention this in the manuscript (lines 166-167)

C: 174 – specify “plant height” here (rather than size)
Re: Done

C: 181 – “Fourteen of the populations…”
Re: Done

C: 182 – were the roadside populations natural or planted?

Re: All the populations were natural. We now highlight this in line 184.

C: 183 – did the use of only reproductive plants introduce a bias?

Re: By focusing on the reproductive plants, we focused on those individuals that were well adapted to the environment. Inclusion of seedlings may have biased the results towards young individuals that may not survive to maturity and thus give inaccurate estimates of the effect of the endophyte on plant distribution with respect to salinity. As salinity was a major determinant of mortality of the young plants in our common garden, including them would have introduced a bias towards plants with high propagule pressure. We now mention this consideration in the methods (lines 187-189).

C: 185-6 – what proportion of the plants were >10m from the shoreline?

Re: 103 out of the 230 plants were >10 m from the shoreline. We now point this out in the methods (lines 192-193).

C: 214 – replace “treatment” with “exposure” here?

Re: Thank you for this suggestion. Exposure is indeed the correct word here and we have made the change as suggested (line 221).

C: 268-70 and 352-4 – height may not be a reliable indicator of overall plant size or performance when comparing plants with and without competition, since the light environment can affect growth habit.  Interestingly it appears there may have been an interaction of endophyte infection and plant growth habit.  Figure 2 shows that  endophyte-infected plants were actually taller under competition (without salt) than they were under benign conditions, whereas endophyte-free plants were similarly tall under both conditions.  This suggests that endophytes were inducing a more sprawling growth habit under benign conditions (i.e., in the open).  Endophytes also affected chlorophyll content, supporting the possibility that they may alter plant response to light.

Re: This is a very good point, and possibly the reason we did not see similar effects in shoot biomass of either first or second year. We have now included these considerations in the discussion (lines 367-368).

C: 290-91 – If many plants were >10m from shore, average distance to shore may not be the most meaningful measure to use here.  Should these data be presented as “> 10 m” vs “< 10 m”, with average distance calculated for the second group?

Re: The reviewer is right that this is not reflecting the true distances to shore as many plants were indeed further than 10 m away from the shore. While it does not reflect the reality, it does present the distances we tested in the model, and therefore we would like to continue to present these means. To address the reviewer’s comment, we have included the proportions of plants that were 10 or more meters away from the shore as well as average distances for under 10m (lines 303-304).

C: 304 – “…only in the salt…”
Re: Done

C: 347 – “…symbioses improve the…”
Re: Done